# AgRP neuron cis-regulatory analysis across hunger states reveals that IRF3 mediates leptin's acute effects

Frankie D. Heyward [1,2,3,4,5] ✉, Nan Liu [6,7,8], Christopher Jacobs [1,2], Natalia L. S. Machado [3,9], Rachael Ivison [1,2], Aykut Uner [1,3,4], Harini Srinivasan[1,2], Suraj J. Patel[1,10,11], Anton Gulko[1], Tyler Sermersheim[1], Linus Tsai [1,2,3] & Evan D. Rosen [1,2,3] ✉

AgRP neurons in the arcuate nucleus of the hypothalamus (ARC) coordinate homeostatic changes in appetite associated with fluctuations in food availability and leptin signaling. Identifying the relevant transcriptional regulatory pathways in these neurons has been a priority, yet such attempts have been stymied due to their low abundance and the rich cellular diversity of the ARC. Here we generated AgRP neuron-specific transcriptomic and chromatin accessibility profiles from male mice during three distinct hunger states of satiety, fasting-induced hunger, and leptin-induced hunger suppression. Cis-regulatory analysis of these integrated datasets enabled the identification of 18 putative hunger-promoting and 29 putative hunger-suppressing transcriptional regulators in AgRP neurons, 16 of which were predicted to be transcriptional effectors of leptin. Within our dataset, Interferon regulatory factor 3 (IRF3) emerged as a leading candidate mediator of leptin-induced hunger-suppression. Measures of IRF3 activation in vitro and in vivo reveal an increase in IRF3 nuclear occupancy following leptin administration. Finally, gain- and loss-of-function experiments in vivo confirm the role of IRF3 in mediating the acute satiety-evoking effects of leptin in AgRP neurons. Thus, our findings identify IRF3 as a key mediator of the acute hunger-suppressing effects of leptin in AgRP neurons.

Increasing the granularity with which we understand the homeostatic regulation of energy balance (i.e., food intake and energy expenditure) will represent a major step towards the ultimate goal of treating obesity. It has long been recognized that the arcuate nucleus (ARC) of the hypothalamus is indispensable for maintaining energy homeostasis[1]. Amid the rich diversity of cell types within the ARC, AgRP neurons play a critical role in coordinating the physiological processes needed to maintain energy homeostasis in the face of changing energy availability.

[1]Division of Endocrinology, Diabetes, and Metabolism, Beth Israel Deaconess Medical Center, Boston, MA, USA. [2]Broad Institute of MIT and Harvard, Cambridge, MA, USA. [3]Harvard Medical School, Boston, MA, USA. [4]Center for Hypothalamic Research, Department of Internal Medicine, UT Southwestern Medical Center, Dallas, TX, USA. [5]Department of Neuroscience, UT Southwestern Medical Center, Dallas, TX, USA. [6]Cancer and Blood Disorders Center, Dana-Farber Cancer Institute and Boston Children's Hospital, Boston, MA, USA. [7]Bone Marrow Transplantation Center of the First Affiliated Hospital, Zhejiang University School of Medicine, Hangzhou, China. [8]Liangzhu Laboratory, Zhejiang University, Hangzhou, China. [9]Department of Neurology, Beth Israel Deaconess Medical Center, Boston, MA, USA. [10]Division of Gastroenterology & Hepatology, UT Southwestern Medical Center, Dallas, TX, USA. [11]Center for Human Nutrition and Department of Internal Medicine, UT Southwestern Medical, Center, Dallas, TX, USA. ✉e-mail: frankie.heyward@UTSouthwestern.edu; erosen@bidmc.harvard.edu

AgRP neurons are activated during states of negative energy balance (i.e., fasting) and selective chemogenetic and optogenetic activation of these neurons drives mice to become markedly hyperphagic[1]. Conversely, post-developmental genetic deletion of AgRP neurons renders mice profoundly hypophagic, and during periods of simulated energy repletion, the satiety-evoking adipokine leptin has been shown to decrease the firing rate of AgRP neurons[2,3]. Finally, CRISPR-Cas9-mediated genetic ablation of the leptin receptor from adult mouse AgRP neurons results in profound obesity that is similar to that exhibited by *db/db* whole-body leptin receptor null mice, indicating that leptin's anti-obesity effects can be mediated via its direct effects on AgRP neurons[4]. Given these findings, much effort has been directed towards elucidating the transcriptional programs that allow AgRP neurons to respond appropriately to changes in energy availability.

Distinct alterations in the transcriptional profiles of AgRP neurons have been noted in response to fasting, including genes encoding neurotransmitter receptors, ion channels, and secreted proteins[5]. These transcriptional changes are suspected to underlie the requisite cell type-specific alterations in neuronal excitability that enable AgRP neurons to influence downstream satiety signaling circuits.

Transcriptional regulation is a key mechanism by which leptin confers its effects on neurons. Mice with a pan-neuronal deletion of signal transducer and activator of transcription 3 (STAT3), the best-known transcriptional effector of leptin action, exhibit profound obesity which is similar to that seen in leptin receptor null mice[6-8]. Moreover, mice devoid of STAT3 specifically in AgRP neurons exhibit an increase in body weight compared to control mice[9]. Activating transcription factor 3 has also been shown to mediate some of the effects of leptin in vivo[10]. Additional evidence of a transcriptional basis for leptin comes from the observation that leptin's suppressive effects on AgRP neuronal activity occur gradually over a 3-hour time course, a finding that implicates the involvement of transcriptional regulation[3].

Previous studies have characterized the transcriptional events that occur in all leptin-receptor-expressing cells in the hypothalamus in response to leptin[10,11]. More recently, one group assessed AgRP neuron-specific transcriptional changes in response to fasting and leptin-administration, observing 33 leptin-induced transcriptional changes that were linked to biological processes suspected to facilitate behavioral changes in response to fluctuations in energy availability[12]. Another group generated a chromatin accessibility landscape of the broad population of leptin receptor-expressing cell types in the brain, yet no bioinformatic assessment of leptin-induced changes in the epigenetic landscape was offered[13]. To date, we lack a rich and comprehensive understanding of the transcriptional and transcriptional regulatory changes that occur in AgRP neurons in response to leptin.

High throughput DNA sequencing techniques that generate genome-wide profiles of open chromatin regions (OCRs) have been used to infer the identity of transcription factors (TFs) that govern the cellular identity and activity state of a variety of cell-types[14-18]. However, applying this approach to study transcriptional regulation within purified populations of AgRP neurons has been challenging for two reasons. The first has to do with the rich cellular heterogeneity within the ARC, where over 50 cell-types have been identified, which necessitates the isolation of these cells prior to their being studied[11]. The second is owed to the relatively low abundance of mouse AgRP neurons (~9000 per mouse) ([19] compared to the cellular input typically required for traditional deeply sequenced genomic profiling techniques ($\geq$ 100,000). To surmount these two obstacles, we developed mouse lines in which AgRP neurons express tagged ribosomes and nuclei, thereby enabling the generation of transcriptomic and epigenomic profiles from as few as 10,000 cells[20]. This enabled us to assess the transcriptomic and cis-regulatory element landscape of AgRP neurons during opposing states of energy availability as a means of identifying additional transcriptional pathways that underlie the distinct cellular states exhibited by these neurons during periods of hunger and satiety.

## Results

### Validating a transgenic tool for obtaining AgRP neuron-specific RNA-seq profiles

We previously developed a mouse model capable of providing cell type-specific transcriptional and epigenomic profiles from complex tissues, which we call NuTRAP (Nuclear tagging and Translating Ribosome Affinity Purification)[20]. NuTRAP mice possess a loxP-stop-loxP sequence immediately upstream of a cassette that expresses a GFP-tagged L10a ribosomal subunit as well as mCherry-tagged RAN-GAP, which localizes to the nuclear membrane. Nuclei can be isolated by flow-sorting for either mCherry or GFP, as the GFP-tagged L10a ribosomal subunit is enriched in the nucleolus, thereby allowing for nuclear sorting. Crossing NuTRAP mice with either AgRP-IRES-Cre or POMC-IRES-Cre mice generates mice with tagged ribosomes or nuclei selectively within AgRP or POMC neurons of the ARC (here referred to as NuTRAP^AgRP or NuTRAP^Pomc). Using this approach, tagged ribosomes can be isolated via affinity purification for RNA-seq, and tagged nuclei can be isolated via fluorescence-activated nuclear sorting (FANS) for ATAC-seq. We confirmed the viability of using NuTRAP^AgRP mice to generate cell-type specific transcriptional profiles by observing distinct gene expression patterns for GFP+ (AgRP neuron positive) and whole sample input ribosomes isolated from NuTRAP^AgRP mice using qPCR. Canonical AgRP neuron genes were enriched, including *Agrp* and *Npy*, whereas genes known to be de-enriched in AgRP neurons were found at reduced levels, such as *Pomc*, *Gfap*, *Mobp* (Fig. S1a). We also observed that TRAP performed on four pooled ARCs resulted in greater enrichment for AgRP neuron markers than that seen with a smaller number of ARCs (Fig. S1A). To overcome the observed minimal input requirements for, and the reliability of, TRAP we opted to use four pooled ARCs for the remainder of our TRAP studies. Next, RNA-seq was used to validate and extend our qPCR data. Again, as anticipated, the canonical marker genes *AgRP* and *Npy* were enriched in our AgRP neuronal population (Fig. S1b). We also observed enrichment for other genes known to be expressed in these neurons, including *Corin*, *Otp*, *Ghsr*, and *Acvr1c* (Supplementary Data 1) (Henry et al., 2017). Notably, the *Serpina3* family of genes was markedly enriched in AgRP neurons, including *Serpina3c*, the third-most enriched transcript, along with *Serpina3i*, *Serpina3n*, *Serpina3g*, and *Serpina3h*, suggesting that this family might play an important role in the regulation AgRP neuron biology. *Lepr*, encoding the leptin receptor, was also significantly enriched in AgRP neurons. We then compared the transcriptomic profiles of NuTRAP^AgRP and NuTRAP^Pomc GFP + TRAP samples. POMC neuron marker genes *Pomc* and *Cartpt* were among the most POMC neuron-enriched genes, along with *Prdm12*, which is indispensable for the expression of *Pomc* (Fig. S1c)[21]. Based on these results we were confident that we could isolate and transcriptionally profile a purified population of AgRP neurons.

### Obtaining AgRP neuron-specific transcriptional profiles in response to fasting and leptin

We next sought to assess transcriptional profiles of AgRP neurons during various states of energy balance using male NuTRAP^AgRP mice that were fed, fasted and leptin-treated (*n* = 3-4 samples per experimental condition, 4 pooled mouse ARCs per sample). Mice were either provided food *ad libitum* or were fasted overnight and then injected with either leptin (5 mg/kg i.p.) or vehicle; three hours later mice were sacrificed and the ARC dissected out for further study (Fig. 1a). The dose of leptin chosen was confirmed to activate leptin-signaling-mediated activation of STAT3 (assessed by phospho-STAT3) in the ARC, with substantial co-localization within the cell body of AgRP neurons (Fig. 1b). We identified 527 and 295 genes that were induced and repressed by fasting in AgRP neurons, respectively (Fig. 1c).

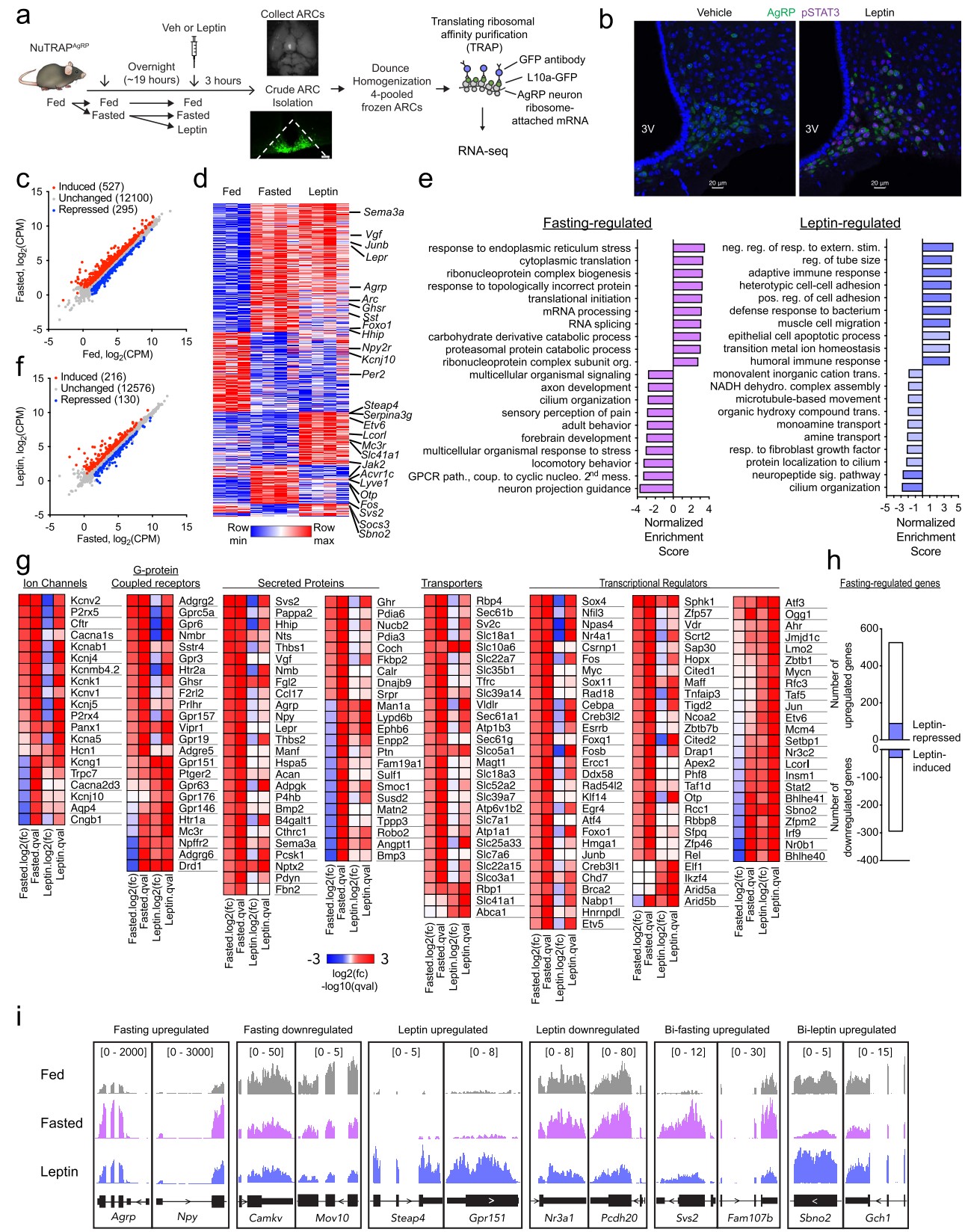

Among the fasting-induced genes were *AgRP*, *Npy*, and *Vgf*, all shown previously to be affected by food withdrawal (Campbell et al., 2017; Henry et al., 2015). Other genes exhibiting increased expression with fasting include *Acvr1c*, *Lyve1*, *Svs2* (Fig. 1d, Supplementary Data 2). WebGestalt (WEB-based Gene SeT AnaLysis Toolkit) gene set enrichment analysis (GSEA) for gene ontology (GO) biological process revealed fasting-induced transcriptional programs that were associated with various molecular functions including, "response to endoplasmic reticulum stress" and "cytoplasmic translation", and "ribonucleoprotein complex biogenesis", among others (Fig. 1e).

**Fig. 1 | Transcriptomic profiles of AgRP neurons during opposing states of energy balance. a** Schematic of the experimental design. NuTRAP^AgRP mice were fed, then a subset were fasted overnight (-19 hr), following which mice were treated with IP vehicle or leptin (5 mg/kg). After 3 hours, mice were euthanized and ARCs were isolated, frozen, and homogenized according to the TRAP protocol. **b** Representative immunofluorescence image of ARC revealing induction of pSTAT3 45-minutes following leptin administration (right) compared to vehicle treatment (left); repeated independently 2 times. **c** Scatter plot showing comparing expression of genes within AgRP neurons from fasted versus fed littermates (n = 4, 3) [fold change >0.5 up (red) or down (blue), false discovery rate (FDR) < 0.05]; CPM, counts per million. The number of genes in a given expression profile is in parentheses. **d** Heatmap of differentially expressed genes in fed, fasted, and leptin-treated mice. **e** WebGestalt (WEB-based Gene SeT AnaLysis Toolkit) gene set enrichment analysis (GSEA) for geneontology biological process in either fasting-regulated (left, induced and repressed), or leptin-regulated (right, induced and repressed) transcriptional pattern categories with Normalized Enrichment Scores. All bars have an FDR < 0.05 except those that are light blue in the leptin-regulated

condition **f** Scatter plot showing regulated genes within AgRP neurons from leptin-treated versus fasted littermates (n = 4, 4) (fold change > 0.5 up (red) or down (blue), FDR < 0.05). CPM, counts per million. FDR-adjusted p-values with a threshold of 0.05 were generated using the Benjamini-Hochberg method to account for multiple comparisons. The number of genes in a given expression profile is in parentheses. **g** Heatmaps of selected gene categories. **h** Top: bar graphs showing the number of fasted up-regulated genes (N = 527) that are also down-regulated with leptin (blue bars, N = 90). Bottom: bar graphs showing the number of fasted down-regulated genes (N = 295) that are also up-regulated with leptin (blue bars, N = 30). **i** Genome browser views (in Integrative Genomics Viewer (IGV)) of RNA-seq tracks of representative fasting-upregulated, fasting-downregulated, leptin-upregulated, leptin-downregulated, bidirectional fasting upregulated (Bi-fasting upregulated), and bidirectional leptin upregulated (Bi-leptin upregulated) genes. The data from TRAP-seq experiments was generated from three independent biological replicates for fed mice, four independent biological replicates for fasted mice, and four independent biological replicates for leptin-treated mice.

These findings recapitulated our earlier work, and that of another group[5,11]. Within our dataset exist fasting-induced genes that are categorized as ion channels (e.g., *Kcnv2*, *P2rx5*, *Cftr*), G-protein coupled receptors (e.g., *Adgrg2*, *Gprc5a*, *Gpr6*), secreted proteins (e.g., *Agrp*, *Svs2*, *Vgf*), transporters (e.g., *Rbp4*, *Sec61b*, *Sc2c*), and transcriptional regulators (e.g., *Sox4*, *Nfil3*, and *Npas2*; Fig. 1g). Unsurprisingly, many transcripts encoding classic immediate early genes (IEGs) were upregulated in AgRP neurons during fasting (e.g., *Nr4a1*, *Fos*, *Myc*, *Fosb*, *Egr1*, *Junb*), potentially driven by the sustained increase in neuronal firing that occurs throughout the course of food deprivation[22].

In leptin-treated mice, we observed the induction of 216, and repression of 130, genes in AgRP neurons (Fig. 1f). *Socs3*, a component of the leptin signaling negative feedback loop, was significantly upregulated in response to leptin-treatment. Notably, leptin-induced transcriptional programs were associated with biological processes including "negative regulation of response to external stimulus", "regulation of tube size", and "adaptive immune response", among others (Fig. 1e). Leptin-induced genes included those encoding ion channels (e.g., *Trpv6*, *Kcng1*), G-protein coupled receptors (e.g., *Drd1*, *Adgrg6*, *Gpr151*), secreted proteins (e.g., *Man1a*, Lypd6b, Ephb6), transporters (e.g., *Rbp4*, *Slc41a1*, *Abca1*), and transcriptional regulators (e.g., *Stat2*, *Bhlhe41*, and *Sbno2*)(Fig. 1g).

We were particularly interested in identifying "bidirectional" transcriptional changes in response to fasting and leptin (i.e., genes induced by fasting and repressed by leptin, or vice versa). This analysis revealed 120 genes exhibiting a bidirectional expression pattern, including 90 genes induced by fasting and repressed by leptin (e.g., *Acvr1c*, *Gpr157*, *Lyve1*, and *Otp)*, and 30 repressed by fasting and induced in response to leptin (e.g., *Sbno2*, *Serpina3i*, *Drd1*, *Man2a1*) (Figs. 1h, 1i). Expression of this set of 30 genes is reduced in response to fasting, a state that has classically been associated with a marked decline in serum leptin levels and leptin signaling, while their expression is increased in response to the administration of exogenous leptin in the fasted state[23]. Given the apparent bi-directional leptin sensitivity of these 30 genes, it is conceivable that leptin depletion-driven declines in the expression of these genes may mediate the adaptive physiological response to starvation attributed to the loss of leptin in the fasted state[23]. We also performed TRAP-seq on fed, fasted and leptin-treated NuTRAP^Pomc samples and observed relatively few significantly changed genes in response to fasting, a finding that recapitulated earlier work, and prompted us to focus on AgRP neurons for the remainder of our studies (Fig. S1d)[5].

### Validating a transgenic tool for obtaining AgRP neuron-specific ATAC-seq profiles

Next, we confirmed the feasibility of using NuTRAP mice to assess chromatin state in AgRP neurons. Using pooled male NuTRAP^AgRP

mouse ARCs, we isolated and sorted 10,000 nuclei, and then performed Assay for Transposase Accessible Chromatin with high-throughput sequencing (ATAC-seq) for AgRP positive (GFP + ) and negative (GFP-) populations from the same pooled samples (Fig. 2a, b). We detected 15,530 and 9029 closed chromatin regions that are specifically enriched in AgRP positive neurons (Fig. S2a), and principal components analysis (PCA) showed a clear distinction between the two populations in terms of their respective chromatin states (Fig. S2b). We next examined the chromatin state of AgRP neurons at key genes expressed in AgRP neurons, compared to that of the non-AgRP neuron population, and observed that AgRP neurons exhibited a selective enrichment of open chromatin regions (OCRs) at the promoters and enhancers of *Agrp*, *Npy*, and *Corin*, among others (Fig. S2c). We also observed multiple OCRs upstream of *Lepr* in AgRP neurons (Fig. S2d). Interestingly, these *Lepr*-associated OCRs are distinct from those observed in hepatocytes, suggesting that putative enhancer elements for *Lepr* may be developmentally divergent. We also detected OCR de-enrichment at the promoter of *Slc17a6*, which encodes the protein vesicular glutamate transporter 2 (VGlut2), reflecting the fact that AgRP neurons are gabaergic, rather than glutamatergic (Fig. S2c). Moreover, de-enrichment of OCRs was noted at non-neuronal marker genes, such as those for oligodendrocytes (e.g., *Olig1* and *Olig2*) and tanycytes (e.g., *Sox2*), supporting the utility and fidelity of our method for obtaining low-input AgRP neuron-specific ATAC-seq profiles (Fig. S2c).

### AgRP neuron-specific ATAC-seq profiles in response to fasting and leptin

We next performed ATAC-seq on AgRP neurons from fed, fasted, and leptin-treated male NuTRAP^AgRP mice. Of the 90,196 called ATAC-seq peaks in our dataset, we detected 18,294 fasted-opened and 8189 fasted-closed regions, and 2134 leptin-opened and 941 leptin-closed regions of chromatin (Fig. 2c, d). We also detected changes in chromatin accessibility that were bidirectional (i.e., reciprocally regulated in fasted vs. leptin-treated) or dually enriched (i.e., open in response to both fasting and leptin-treatment) (Fig. 2c). The majority of differentially expressed ATAC peaks were found within intergenic (36–45%) and intronic (43-50%) regions, while only 3-11% of ATAC-peaks were found within gene promoter regions, defined as regions <1000 bps upstream of the transcriptional start site (TSS) (Fig. 2e), a changed genomic distribution of changed chromatin accessibility previously exhibited by activated neurons (Su et al., 2017). We observed areas of chromatin that became more accessible with fasting (i.e., fasted-opened) near various genes known to be enriched in response to fasting in AgRP neurons, including *Npy* (Fig. 2f, g). Moreover, we also observed areas of chromatin that became less accessible with fasting (i.e., fasted-closed), as well as those that became more or less accessible with leptin (i.e., leptin-opened and leptin-closed) (Fig. 2g).

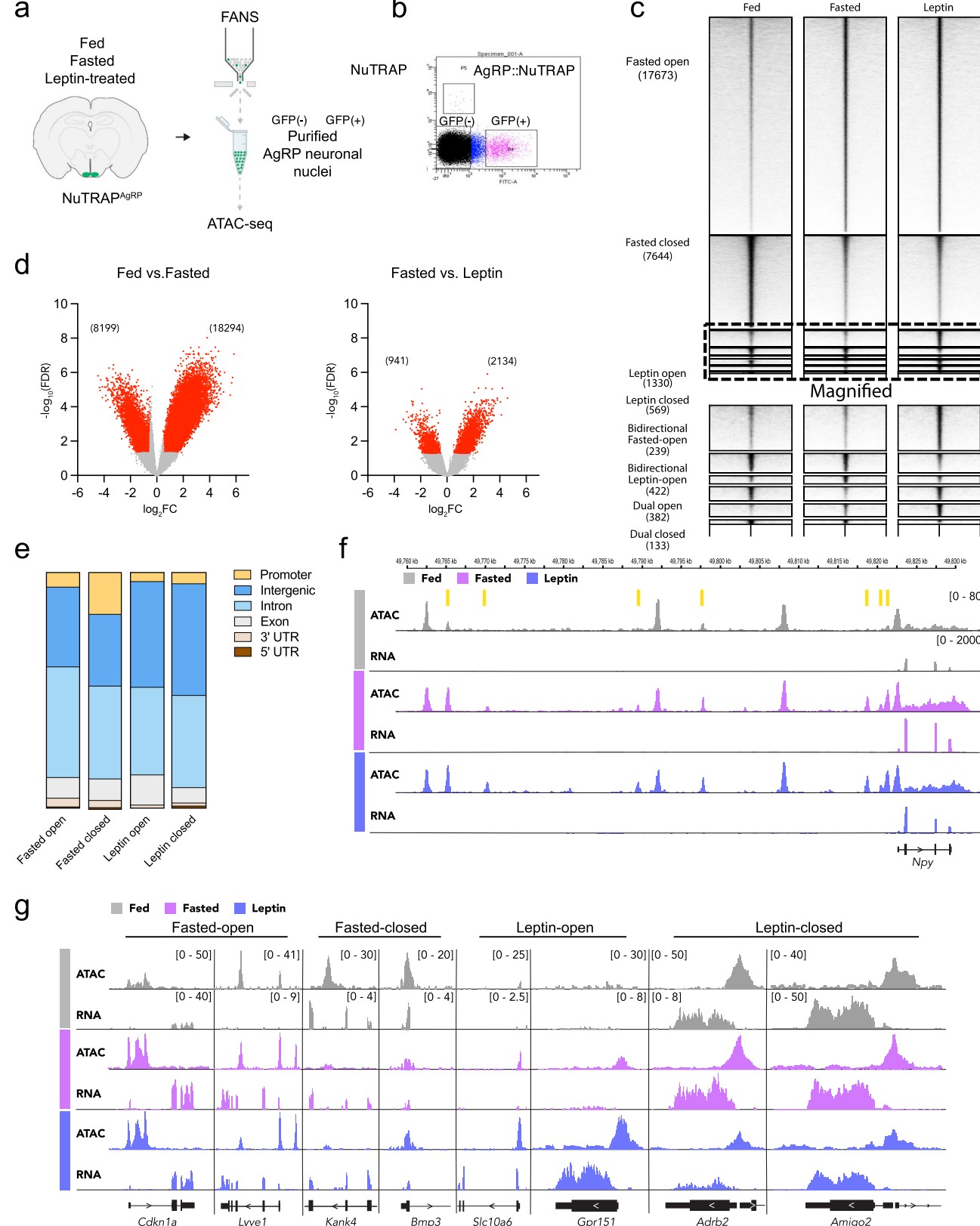

## Uncovering transcriptional regulators that control AgRP neuron biology

Having generated an extensive atlas of dynamic chromatin regions in AgRP neurons in response to fasting and leptin treatment, we next sought to determine which TFs might bind to these regions. To accomplish this we performed Analysis of Motif Enrichment (AME), a program of the MEME suite, on chromatin regions with a high likelihood of impacting fasting- and leptin-induced transcriptional changes (Fig. 3a)[24]. To accomplish this, we filtered ATAC-seq peaks to retain those within ± 200 kb of a TSS corresponding to a gene whose expression is concordantly regulated (e.g., fasting-induced, gained-open ATAC-seq peaks near fasting-induced genes; hereafter called

**Fig. 2 | Alterations in energy availability impact the chromatin landscape of AgRP neurons. a** Experimental schematic. ARCs from fed, fasted, and leptin-treated NuTRAP^AgRP mice were isolated followed by nuclear isolation. 10,648 ± 1489 nuclei, from 4 mice, were pooled for each condition (fed, fasted, leptin-treated) before being subjected to ATAC-seq. **b** FACS plot of single nuclei with FITC (GFP) fluorescence on the x-axis and mCherry fluorescence on the y-axis, with filtering gates showing discrete populations for NuTRAP^AgRP GFP+ and GFP- nuclei. **c** Heatmap showing differentially expressed OCRs for each of the treatment conditions clustered into one of eight patterns (rows). Amplitude of each peak center (±5 kb) is represented in black as indicated. **d** Volcano plots showing differentially expressed OCRs in fasted vs. fed mice (left, n = 2, 2) or leptin vs. fasted mice (right, n = 2, 2). Red dots correspond to significantly different gained-open and gained-closed regions [fold change >0.5 up (red) or down (red), FDR < 0.05]; CPM, counts per million. Peaks with a CPM < 1 were not included in volcano plot. FDR-adjusted p-values with a threshold of 0.05 were generated using the Benjamini-Hochberg method to account for multiple comparisons. **e** Bar graphs revealing the genomic features of fasted-opened, fasted-closed, leptin-opened, leptin-closed ATAC-seq peaks. **f** Genome browser views (IGV) of ATAC-seq peaks within the vicinity of the *Npy* gene. Yellow bars correspond to fasted-opened ATAC-peaks. **g** Genome browser views (IGV) of representative Fasted-opened, Fasted-closed, Leptin-opened, and Leptin-closed ATAC-seq peaks near the indicated genes. The data from ATAC-seq experiments was generated from two independent biological replicates for fed mice, two independent biological replicates for fasted mice, and two independent biological replicates for leptin-treated mice.

concordant fasted-opened; Fig. 3a, S3a). We observed that 96% (p = 1.39 ×£10^{−9}), 78% (p = 1.17 × 10^{−8}), 59% (N.S.), and 51% (N.S.) of fasting-induced, fasting-repressed, leptin-induced, and leptin-repressed genes, respectively, were associated with a concordant peak (Fig. S3b). Thus, up-regulated genes were significantly more likely to be close to an up-regulated peak in the same condition, but this was not true for down-regulated genes. The majority of genes associated with a concordant peak were in fact associated with numerous concordant peaks; most genes were associated with 4-6 peaks for fasting-induced genes, and 1–3 peaks per genes for the other three comparisons (Fig. S3c). We also observed a positive relationship between the number of concordant peaks and the fold-change of the associated gene (Fig. S3d). The number of concordant peaks identified for each of the four comparisons was 2452 fasted-opened, 531 fasted-closed, 203 leptin-opened, and 74 leptin-closed (Fig. 3b).

We next performed motif enrichment analysis on the four distinct sets of concordant peaks (i.e., fasted-opened, fasted-closed, leptin-opened, leptin-closed), while employing 6 different background control peak sets. Thus, in all, 6 separate instances of AME were performed for each peak set. Motifs were considered further only if they were enriched in ≥5 out of 6 AME instances (Fig. S3a, S3e, S3f). We observed that motifs associated with the AP-1 transcription factor family were highly enriched in concordant fasted-opened peaks, including motifs for FOSB, JUNB, ATF3, JUN, and JUND (Fig. 3c, d). This finding was unsurprising given that these activity-dependent TFs have been implicated as being permissive for the physiological changes that enable neurons to increase their firing rate[25]. Reassuringly, STAT3 was the most enriched TF motif in leptin-opened peaks, consistent with its established role as a key mediator of leptin-induced transcriptional regulation (Fig. 3c).

We next set out to develop a prioritized list of TFs that might drive the expression of genes that increase AgRP neuron firing during fasting. We reasoned that such TF motifs would be enriched in concordant fasted-opened and leptin-closed peaks, but not in concordant leptin-opened or fasted-closed peaks. With this approach, we identified 101 TF motifs that were enriched in concordant fasted-opened and leptin-closed peaks (Fig. 3e). Of these motifs, only 4 were not also enriched in concordant leptin-opened and fasted-closed peaks (Fig. 3f). Upon ranking these motifs by their summed motif score, PRGR and ANDR, two transcription factors not previously implicated in the control of AgRP neuron biology, were the most enriched (Fig. 3g). We also identified 14 TF motifs that were exclusively enriched in fasted-opened peaks, with the MAF transcriptional factor family members, MAFB, MAFK and MAFG having been particularly enriched (Figs. 3d, 3h).

Next, we developed a prioritized list of TFs that potentially drive the expression of genes that decrease AgRP neuron firing after leptin treatment. We reasoned that such associated TF motifs would be enriched in the two distinct sets of concordant leptin-opened peaks and fasted-closed peaks, but not fasted-opened or leptin-closed peaks. We identified 127 TF motifs that were enriched in both concordant leptin-opened and fasted-closed peaks (Fig. 3i). Of these, 16 TF motifs were also not enriched in fasted-opened or leptin-closed peaks

(Fig. 3f). We then ranked these 16 TF motifs based on their summed motif scores (summed motif enrichment scores in leptin-opened and fasted-closed peaks) (Fig. 3j). With this approach we found several motifs that corresponded to TFs that have not been linked to the neural control of energy homeostasis, including interferon regulatory factor 3 (IRF3), which had the most significant concordant fasted-closed motif enrichment score, and third most significant summed motif score, behind only STAT3 and ETS2 (Fig. 3i). Moreover, the IRF3 motif enrichment profile was similar to that of STAT3, and distinct from those enriched in fasted-open and leptin-closed areas of chromatin (e.g., PRGR, MAFB) (Fig. 3k).

## Leptin activates neuronal IRF3

*Irf3* mRNA is expressed in AgRP neurons; we saw no change in *Irf3* levels with fasting or leptin treatment (Fig. 4a), which was unsurprising given that *Irf3* is not nutritionally regulated in other cell types[26,27]. We also noted the existence of an OCR near the *Irf3* TSS (Fig. 4a). Additionally, we detected IRF3 protein expression within the ARC of adult male mice using immunofluorescence (Fig. 4b).

In immune cells, adipocytes, and other cells, IRF3 is a key component of the antiviral innate immune system and can be activated by three established pathways: (1) dsRNA → Tlr3/4; (2) dsRNA → MDA5/Rig-1; or (3) dsDNA → cGAS. Upon activation by upstream kinases, IRF3 translocates from the cytosol into the nucleus to activate gene expression[28]. Interestingly, in contrast to other toll-like receptors, Tlr3 was appreciably detected in AgRP neurons, suggesting a potential role for Tlr3 → Irf3 activation in AgRP neurons (Fig. S4a). Moreover, in addition to Tlr3 and its signaling components, the MDA5/Rig1 pathway is also expressed, all of which suggest a potential route and role for IRF3 activation in AgRP neurons (Fig. S4b, c). Interestingly, many canonical inflammatory genes (e.g., *Isg15, Rsad2, Oasl1, Ccl5*) which have been linked to IRF3 activation in macrophages, hepatocytes, and other cell types are very lowly, if at all, expressed, in AgRP neurons, either in the presence or absence of leptin. Using ATAC-seq data from hepatocytes as a comparator, we note that various known IRF3 target genes that are expressed in hepatocytes but not in AgRP neurons (e.g., *Rsad2, Isg15, Ccl5*, and *Oasl2*) are associated with OCRs in hepatocytes only (Fig. S4d). Thus, we predict that the transcriptional targets of IRF3 in AgRP neurons are distinct from those found in other cell-types.

To determine if IRF3 is activated by leptin, we employed an in vitro model. GT1-7 cells are an immortalized mouse gonadotrophin-releasing hormone (GnRH)-expressing hypothalamic neuronal cell line that are rendered leptin responsive when transfected with the leptin receptor[29]; the leptin-responsivity of these cells was confirmed by detecting an increase in pSTAT3 following leptin-treatment (Fig. 4c, S4e). GT1-7 cells were co-transfected with plasmids expressing the long-form of the mouse leptin receptor, and IRF3-GFP, respectively. These cells were then treated with either vehicle or 100 nM leptin prior to being subjected to live cell imaging. With this approach we observed a pronounced induction in nuclear translocation of IRF3 upon treatment with leptin at the 5-hour time point (Fig. 4b). A similar time course of IRF3 activation has been demonstrated using models of

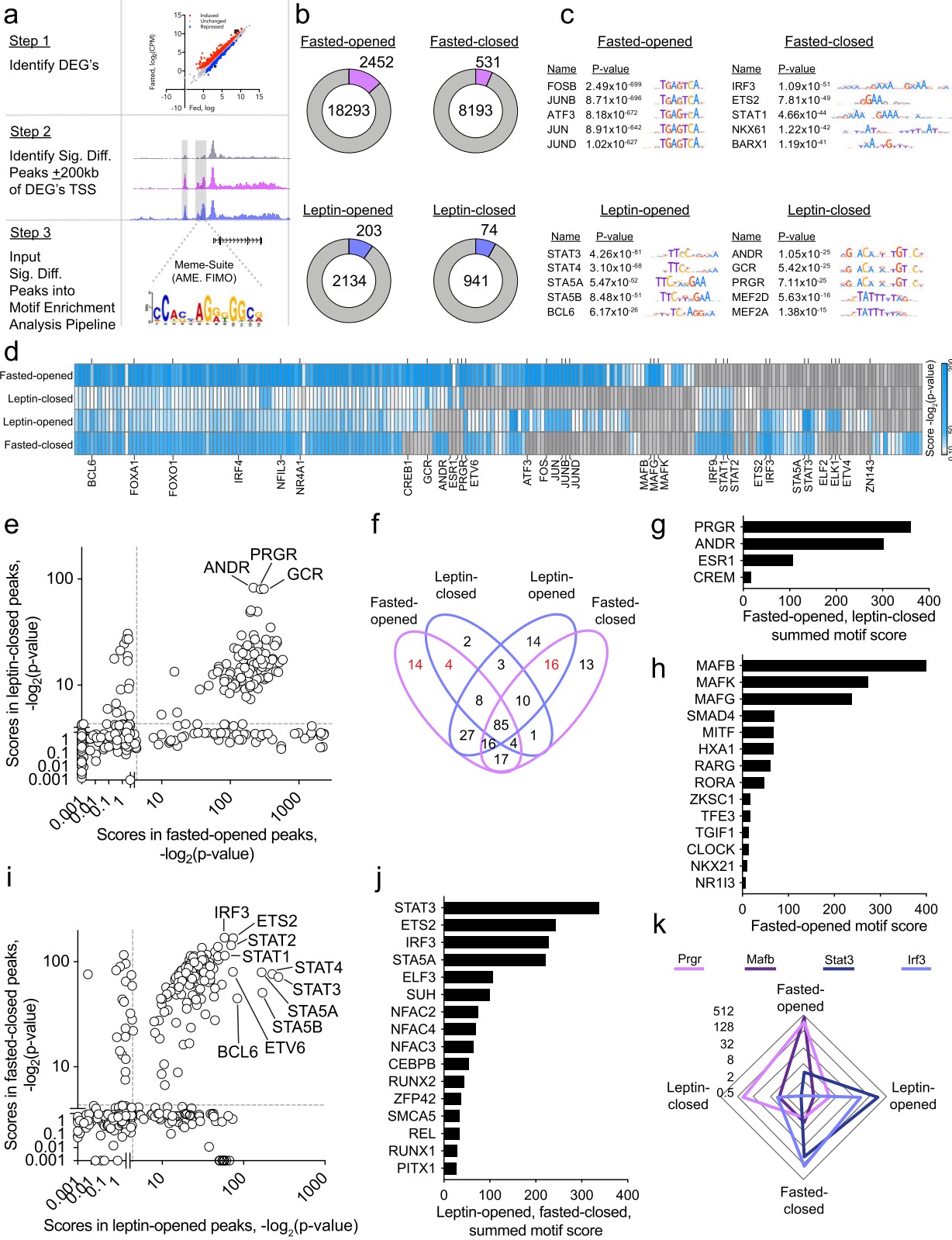

Poly(I:C)-driven IRF3 action HT-29 cells[30]. These results are consistent with a model whereby leptin induces the activation of neuronal IRF3 in a cell-autonomous manner.

To determine whether leptin activates IRF3 in vivo, whole hypothalamus was collected from wild-type male C57BL6/J mice 5 h after treatment with either vehicle or leptin, an approach we confirmed is suitable for detecting leptin-induced nuclear pSTAT3 (Fig. S4f). Isolated nuclei were treated with an IRF3 antibody and fluorescent secondary antibody and subjected to flow cytometry (Fig. 4e). With this approach we detected a significant increase in nuclear IRF3 following leptin-treatment, a finding that demonstrates the ability of leptin to activate IRF3 within the mouse hypothalamus (Fig. 4f).

**Fig. 3 | Integrated transcriptomic and cistromic analysis identifies putative leptin-sensitive TFs in AgRP neurons. a** Schematic showing the broad computational heuristic employed to identify putative pro-satiety TF motifs. Step 1: Identify fasted-induced, fasting-repressed, leptin-induced, and leptin-repressed differentially expressed genes (DEGs). Step 2: Determine significantly different fasted-opened, fasted-closed, leptin-opened, and leptin-closed ATAC-seq peaks, and identify those that are ± 200 kb upstream and downstream of those DEGs identified in Step 1 (e.g., concordant leptin-opened peaks). Step 3: Perform motif enrichment analysis on the peaks identified in Step 2. **b** Pie charts showing the total number of peaks classified as fasted-opened, fasted-closed, leptin-opened or leptin-closed, as well as the subset of concordant peaks (lavender and blue bars). For this analysis, we filtered non-chromosomal features (scaffolds); there were 8 such instances in the fasted vs fed comparison and none in the leptin vs. fasted comparison. **c** TF motifs enriched in concordant fasted-opened, fasted-closed, leptin-opened and leptin-closed ATAC-seq peaks. **d** Heatmap showing motif scores [-log$_2$(p-value)]

concordant fasted-opened, leptin-closed, leptin-opened, and fasted-closed ATAC-seq peak sets. (**e**) Comparison of TF motif enrichment scores for fasted-opened peaks vs. leptin-closed peaks. **f** Venn diagrams showing the number of TF motifs enriched in concordant fasted-opened, leptin-closed, leptin-opened, and/or fasted-closed ATAC-seq peaks. **g** Summed motif enrichment scores from fasted-opened and leptin-closed TF motifs. **h** Motif enrichment scores from fasted-opened TF motifs. **i** Comparison of TF motif enrichment scores for leptin-opened peaks vs. fasted-closed peaks. **j** Summed motif enrichment scores from leptin-opened and fasted-closed TF motifs. **k** Radar plot showing the TF motif enrichment scores (-log$_2$(p-value)) with 4 axis with a log2 scale, all for concordant fasted-opened, leptin-closed, leptin-opened, and fasted-closed ATAC-seq peak sets. Profiles shown are for PRGR, MAFB, STAT3, and IRF3. For AME analysis, the term p-value is the optimal enrichment p-value of the motif according to the statistical test, adjusted for multiple tests using a Bonferroni correction.

## IRF3 mediates leptin-induced satiety in AgRP neurons

We have previously observed that whole body IRF3 knockout mice are hyperphagic on a high-fat diet compared to control mice, a finding that was at the time interpreted as a compensatory mechanism to offset the increased energy expenditure that these mice exhibit[27]. Here we asked whether IRF3 mediates the hunger-suppressive effects of leptin directly. We generated male AgRP-ires-Cre::*Irf3*$^{fl/fl}$ (AgI3KO) mice as a means of deleting IRF3 selectively within AgRP neurons. No change in chow-fed body weight or cumulative food intake was apparent during basal conditions in AgI3KO mice (Fig. S5a, b). We next implemented an established fasting-refeeding paradigm to assess leptin-induced satiety (Fig. 5a). In this paradigm, leptin-treated AgI3KO mice exhibited greater 24-hour cumulative food intake upon refeeding than control mice, indicating blunted sensitivity to leptin (Fig. 5c). Of note, control mice reduced their food consumption by 1.14 gm over 24 hours, while AgI3KO mice reduced their intake by only 0.38 gm over the same time period, a 67% reduction in leptin action. The reduced responsivity to leptin exhibited by AgI3KO mice is unlikely to be due to general hyperphagia, or greater sensitivity to starvation, per se, as no difference was observed between vehicle-treated fasted-refed AgI3KO mice and control mice (Fig. 5b).

IRF3 is activated by viral RNA binding to Toll-like receptor 3 (TLR3), which is enriched in AgRP neurons relative to other TLRs (Fig. S4a). The synthetic (TLR3) ligand polyinosinic:polycytidylic acid (poly(I:C)), a known activator of IRF3, induces reduced locomotion, anorexia, and early phase hyperthermia followed by a late phase (8-16 hours) hypothermia[31,32]. Interestingly, leptin has been shown to produce not only anorexia, but also an alteration in core body temperature that resembles a sickness response[33]. Thus, given the high probability that AgRP neurons are a component of the neural ensemble involved in the sickness response, we asked whether IRF3$^{AgRP}$ may mediate behavioral and physiologic responses to leptin that are commonly observed during a sickness response (i.e., reduced locomotion and derangements in thermoregulation). To this end, telemetry probes were implanted in AgI3KO and control mice to measure core body temperature and locomotor activity (LMA) in two-dimensional space, and vehicle or leptin was infused directly into the ARC (Fig. 5d). Leptin-treated control mice exhibited a decrease in late-phase (8-16 hours) core body temperature (*P = .0402) (Fig. 5e,£f). Leptin-treated AgI3KO mice exhibited a late-phase (8-16 hour) core body temperature that was significantly higher than that of control mice (Fig. 5e, h, S5c), despite no difference in core body temperature between vehicle-treated control and AgI3KO mice (Fig. 5e, g, S5c). Moreover, leptin significantly reduced LMA in control mice (*P = .0172), while there was a trend ($p = 0.07$) towards increased LMA in AgI3KO mice after leptin (Fig. S5d–f). Thus, these findings demonstrate the blunted effects of leptin on mice devoid of IRF3 in AgRP neurons.

To assess whether IRF3 activation in AgRP neurons is sufficient to suppress food intake, we utilized a gain-of-function mouse model in

which Ser388/Ser390 of IRF3 are mutated to phospho-mimetic Asp residues (S → D), creating a constitutively active allele that is expressed in a Cre-dependent manner (the allele and mouse are hereafter referred to as IRF3-2D)[34]. These IRF3-2D mice were crossed to AgRP-IRES-Cre mice, thereby generating AgIRF3-2D mice which expressed the constitutively active version of IRF3 in their AgRP neurons (Fig. S5f). In vivo flow cytometry analysis revealed an increase in nuclear IRF3 within hypothalamic nuclei of AgI3-2D mice (Fig. S5g). As predicted, fasted, vehicle-treated, male AgI3-2D mice have suppressed food intake following a period of fasting, thus mimicking that of control mice treated with leptin (Fig. 5g, h).

## Discussion

While the ability of leptin to evoke prolonged changes in food consumption has largely been attributed to its engagement of the transcriptional effector STAT3, we lack a complete roster of leptin-sensitive TFs in any cell type[3]. Converging findings indicate a role for leptin-sensitive, STAT3-independent transcriptional effectors. For instance, the dose of leptin required to suppress food intake in rats is much higher than the dose required to produce maximal STAT3 activation (i.e., phospho-STAT3) within the ARC, suggesting that leptin-mediated activation of pSTAT3 is not sufficient to yield a full satiety-evoking effect[35]. Moreover, mice lacking STAT3, or the tyrosine residue of LepR that recruits STAT3 (Tyr$_{1138}$), are less obese than mice lacking either leptin (*ob/ob*) or the leptin receptor (*db/db*), and leptin signaling remains partially intact in these mice[36]. These results suggest the existence of a STAT3-independent arm of LepR signaling. Interestingly, LepRb sequences between residues 921 and 960 have been implicated as indispensable for the STAT3-independent component of leptin's metabolic actions[36]. Our data indicates a role for IRF3 as a transcriptional mediator of leptin action in AgRP neurons.

Our simultaneous examination of transcriptomic and chromatin accessibility landscapes during opposing states of hunger and leptin-induced hunger-suppression enabled us to generate a list of candidate TFs that potentially regulate hunger in a leptin-dependent manner. The observation that the IRF3 motif was enriched in concordant leptin-opened peaks suggested that leptin-driven activation of IRF3 may be permissive for the expression of "hunger-suppressing" transcriptional programs in AgRP neurons. Consistent with this model, loss-of-function AgI3KO mice exhibited reduced leptin-induced suppression of hunger during a fasting-refeeding paradigm. Alternatively, or in addition, the finding that the IRF3 motif was enriched in concordant fasted-closed peaks suggested that the loss of IRF3 activity might be permissive for expression of "pro-hunger" transcriptional programs in AgRP neurons in the fasted state. Given the decline in serum leptin levels and leptin signaling during the fasted state, it stands to reason that a decline in IRF3 activity during fasting might be necessary for the establishment of a "pro-hunger" transcriptional state in AgRP neurons. Consistent with this model, gain-of-function AgI3-2D mice exhibited

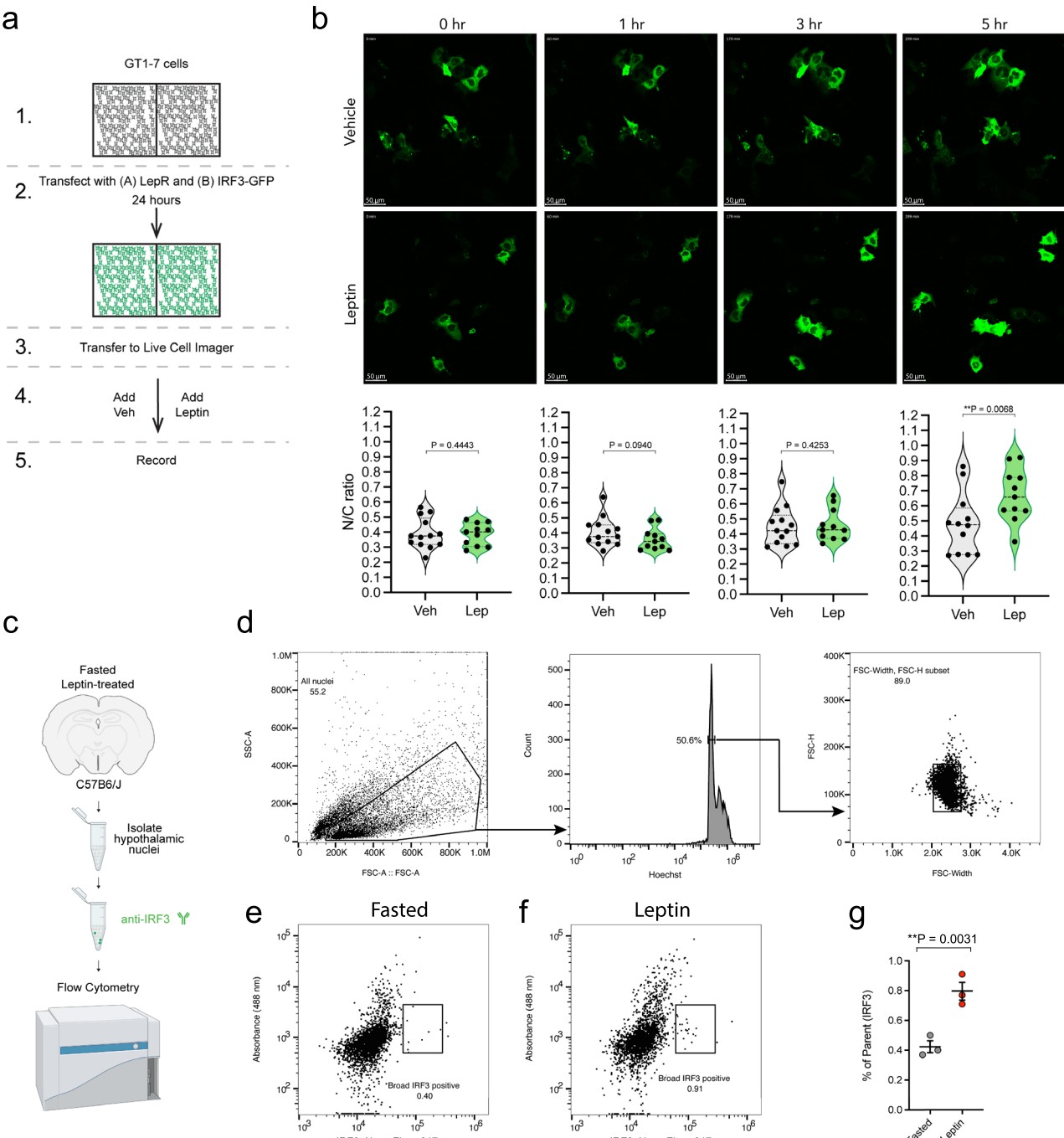

**Fig. 4 | Leptin activates IRF3 in a cell autonomous manner. a** Schematic of in vitro GT1-7 live-cell imaging experiment. **b** Representative images showing the dynamic imaging of GFP-tagged IRF3 in response to vehicle or leptin stimulation in GT1-7 cells expressing both IRF3-GFP and leptin receptor at 0, 1, 3, and 5 hours after stimulation. Quantification of nuclear/cytoplasmic (N/C) GFP signal is shown at each time point. Each dot represents a single tracked cell ($n = 13, 11$). At T = 5 hour ($p = 0.0068$, t = 2.681, df = 22). The violin plots indicate median values (wide dashed middle line), first and third quartiles (narrow dashed lines); this finding recapitulated an earlier observation from a pilot experiment. **c** Schematic of in vivo IRF3 flow cytometry experiment. **d** Flow cytometry gating strategy. Particles smaller than nuclei (black dots) were eliminated with an area plot of forward-scatter (FSC-A) versus side-scatter (SSC-A), with gating for nuclei-sized particles inside the gate (box)[52]. 2 N Hoechst 33342 stained nuclei were positively gated to avoid nuclei doublets. Plots of width versus height, both in the forward-scatter channel were used to further exclude aggregates of two or more nuclei. **e**, **f** Two representative scatter plots of IRF3-Alexa Fluor647 primary and secondary staining (x-axis) and empty 488 channel (y-axis) were used to identify the IRF3-positive population for the fasted (**e**) and leptin-treated (**f**) conditions. **g** Results of the fasted and leptin % of Parent IRF3 signal comparison. Results are presented as mean values +/- SEM and analyzed by a one-tailed student t-test ($p = 0.0031$, t = 5.251, df = 4). This finding was replicated once. The flow cytometry results were generated from three independent biological replicates for fasted mice, and three independent biological replicates for leptin-treated mice.

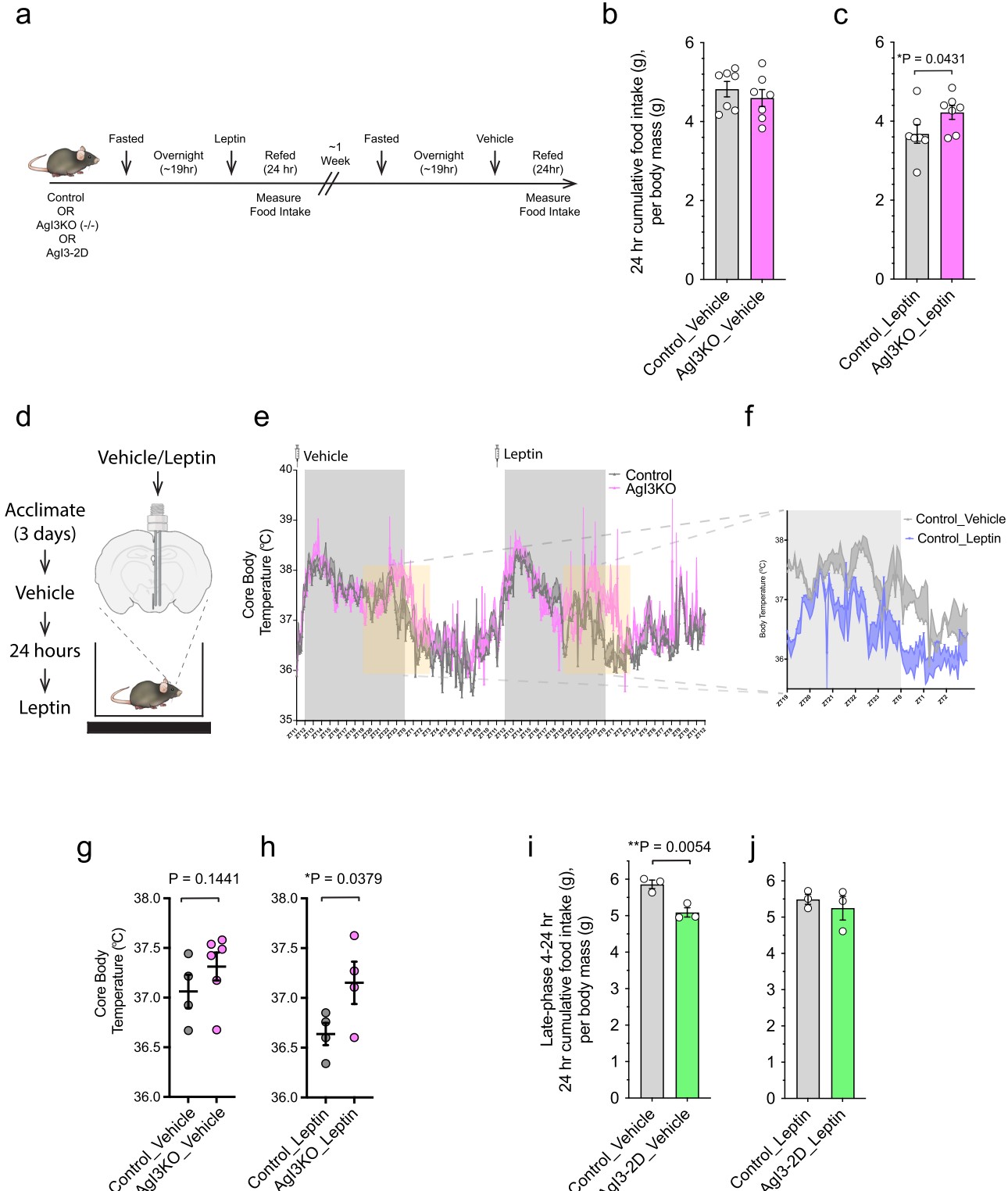

reduced hunger when tested in the fasting-refeeding paradigm. Given these findings, we propose that IRF3 represents a leptin-sensitive TF that bidirectionally regulates hunger, with the loss of leptin-mediated IRF3 signaling in the fasted state representing a critical "pro-hunger" event, and the resurgence in leptin-mediated IRF3 signaling in the leptin-treated state representing a key "hunger-suppressing" event. IRF3 has been chiefly studied as a core component of the innate immune antiviral response pathway, with viral pathogen-associated molecular patterns (PAMPs) binding to TLR3/4 and activating the

cGAS-STING/TBK1 pathway, resulting in phosphorylation and nuclear translocation of IRF3[37,38]. The signal transduction mechanisms linking leptin receptor to IRF3 remain to be determined, but we note that several of the known upstream activators of IRF3 (e.g., cGAS, STING, and IKKε) are not expressed in AgRP neurons, according to our RNA-seq results. It is interesting to speculate that PI3K-Akt, a known but poorly understood arm of the leptin signaling pathway, may be involved. This hypothesis is supported by data showing that macrophage IRF3 is phosphorylated and activated by Akt, and this activation

**Fig. 5 | IRF3 in AgRP neurons mediates leptin-induced behavioral alterations.**
**a** Schematic of fasting-refeeding experimental paradigm. Control and AgI3KO mice
were fasted overnight, followed by refeeding in the presence of leptin; food intake
was measured over a 24-hour period. After a week, the same mice were subjected to
the same paradigm, except vehicle was administered instead of leptin. **b** 24-hour
cumulative body-weight adjusted food intake measured in fasted-refed, vehicle-
treated, control and AgI3KO mice ($n = 7, 7$). **c** 24-hour cumulative body-weight
adjusted food intake measured in fasted-refed, leptin-treated, control and AgI3KO
mice ($n = 7, 7$). Results are presented as mean values +/- SEM and analyzed by a one-
tailed student t-test ($p = 0.0431$, t = 1.869, df = 12). Food intake study data (Fig. 5b, c)
were generated from seven independent biological replicates for control mice, and
seven independent biological replicates for AgI3KO mice. Each circle denotes a
single mouse. Results are mean ± SEM of each condition and analyzed by a one-
tailed student t-test. **d** Schematic showing the time-course and experimental
paradigm of vehicle- and leptin-treatment telemetry experiments. **e** Core body
temperature measures during a 48-hour period showing control mice that received
vehicle-and leptin-treatment (5 mg/kg) in control mice or AgI3KO mice ($n = 4, 6, 4,$
4). The syringes and transparent yellow boxes illustrate the times of vehicle/leptin
injection and the 8-hour epochs used for analyzes, respectively. **f** Inset image
showing superimposed Control_Vehicle and Control_Leptin core body temperature

plots during their respective second 8-hour recording epochs. **g** Average core body
temperature following vehicle-treatment for control and AgI3KO mice, with each
mouse's mean value during the 8-hour period depicted ($n = 6, 3$). **h** Average core
body temperature following leptin-treatment for control and AgI3KO mice, with
each mouse's mean value during the 8-hour period depicted ($n = 3,3$). Core body
temperature study data in Fig. 5G were generated from four independent biological
replicates for control mice, and six independent biological replicates for AgI3KO
mice. Results are presented as mean values +/- SEM and analyzed by a one-tailed
student t-test ($p = 0.1441$, t = 1.138, df=8). Core body temperature study data in
Fig. 5H were generated from four independent biological replicates for control
mice, and four independent biological replicates for AgI3KO mice. Results are
presented as mean values +/- SEM and analyzed by a one-tailed student t-test
($p = 0.0379$, t = 2.143, df=6) **i** 4-24-hour body weight adjusted cumulative food
intake measured by fasted-refed, vehicle-treated, control and AgI3KO mice ($n = 3,$
3). Results are presented as mean values +/- SEM and analyzed by a one-tailed
student t-test ($p = 0.0054$, t = 4.507, df = 4) **j** 4-24-hour body weight adjusted
cumulative food intake measured by fasted-refed, leptin-treated, control and
AgI3KO mice ($n = 3, 3$). Food intake study data (Figs. 5i, 5j) were generated from
three independent biological replicates for control mice, and three independent
biological replicates for AgI3KO mice. Each circle denotes a single mouse.

is blocked by the PI3K inhibitor wortmannin[39,40]. Future studies will
address potential leptin → PI3K/Akt → IRF3 pathway and its contribu-
tion to leptin-induced satiety.

Mice with IRF3 ablation in AgRP neurons exhibited blunting of
leptin sensitivity, yet did not exhibit either a basal or high-fat diet-
induced increase in body weight. This finding is not unexpected. First,
Cre-loxP deletion of *Lepr* from AgRP neurons results in only a mild
increase in body mass and composition that was primarily driven by
decreased energy expenditure, as no change in basal food intake was
detected[41]. More recently, CRISPR–Cas9-mediated genetic ablation of
*Lepr* in AgRP neurons of adult mice was shown to cause an increased
body weight phenotype that recapitulated that observed with whole-
body *Lepr* null mice, suggesting that deletion of the leptin receptor
early in development results in yet-to-be elucidated compensation that
preserves the function of the core homeostatic regulatory machinery[4].
A similar developmental compensation may contribute to the lack of a
body weight phenotype in our AgI3KO mice. There is known functional
redundancy between IRF3 and IRF7, and IRF7 or another TF (e.g.,
STAT3, STAT1), may be able to compensate for the absence of IRF3[37]. It
should be mentioned that there is precedent for Cre-loxP deletion of a
suspected "pro-satiety" transcription factor abrogating leptin-induced
satiety in the absence of a body weight phenotype, as was observed
with the deletion of ATF3 from leptin receptor-expressing cells (Allison
et al., 2018). Future studies should assess the impact of loss of IRF3 in
adult AgRP neurons using a time-restricted approach. Moreover, our
study was designed to identify transcriptional regulators that influence
AgRP neuronal biology after an acute administration of leptin.
Experiments using lean and obese mice will identify alternative can-
didate TFs that regulate programs underlying obesity.

IRF3 is expressed in most mammalian cells, and some of its tran-
scriptional targets have been identified in macrophages and
adipocytes[27,34,42]. In immune cells, IRF3 is known to drive the expres-
sion of interferon beta (*Ifnb1*) along with a whole host of interferon-
stimulated genes (ISGs), cytokines, and other mediators of the innate
antiviral response pathway[37]. Interestingly, our study revealed that
many canonical IRF3 target genes were not detected in AgRP neurons
during either of the two states in which we predict IRF3 is active (e.g.,
fed and leptin-treated), including *Ifnb1*, *Isg15*, *Isg54*, *Isg56*, *Ccl4*, *Ccl5*,
*Cxcl10*, among others. Thus, IRF3 in AgRP neurons has transcriptional
targets that are distinct compared to those found in other cell types.
Our ATAC-seq data suggests that this may be due to an underlying
chromatin landscape in AgRP neurons that restricts access of IRF3 to
classical target genes. It is also possible that specific interacting pro-
teins direct IRF3 to distinct loci in hepatocytes and immune cells vs.

neurons. The IRF3 cistrome has not been determined in any other CNS
cell types, and so the generalizability of this phenomenon is unclear,
although it should be noted that many transcription factors exhibit
different cistromes in different cell types [57–59].

It is interesting to speculate why a pro-inflammatory transcription
factor might be co-opted for use in energy homeostasis. During illness,
such as after viral infection, appetite is generally suppressed, an effect
known as the sickness response. Polyinosinic:polycytidylic acid
(Poly(I:C)), a synthetic analog of double-stranded RNA and a potent
activator of TLR3, has been shown to mimic a viral infection and is able
to evoke a classic sickness response, including hypophagia and
decreased locomotor activity, when systemically or centrally admi-
nistered to mice[31,32]. IRF3 is a dominant transcriptional effector of TLR3
receptor activation[37], and it is highly probable that IRF3 mediates the
transcriptional programs underlying the prolonged sickness behaviors
caused by poly(I:C) treatment or viral infection. It might therefore
make sense for evolution to converge on the same pathway as an
effector of homeostatic appetite control outside of the context of ill-
ness, such as during fasting or feeding.

Our study has certain limitations. First, all of our experiments
were performed in male mice, and it is possible that results may differ
in females. Second, we used 3 hours-post leptin administration since it
had been previously shown that leptin gradually suppresses the firing
rate of AgRP neurons over the course of 3 h (Beutler et al., 2017). Thus,
we suspected that we would be able to detect discernable transcrip-
tional changes related to the alterations in the firing rate of AgRP
neurons following leptin-treatment at this time point. We may be
missing transient transcriptional changes in various IEGs, or other
transcripts. Third, most of our studies are restricted to AgRP neurons,
and it remains to be seen whether IRF3 has a role downstream of leptin
signaling in other leptin sensitive cell types. Finally, it is unclear whe-
ther our studies inform us about human biology. Variation at the *IRF3*
locus has not been associated with body weight in humans, either in
GWAS studies or in exome sequencing of patients with extreme obe-
sity. Consistent with this, we have not seen a major effect on body
weight in AgI3KO mice. Our data support a role of IRF3 in mediating
the acute effects of leptin on AgRP neuronal activity—it's likely that
other leptin signaling pathways are more important in chronic weight
maintenance.

## Methods
### Mouse models
**Generation of mice.** IRF3-2D mice (Jackson Labs Strain #036261) and
IRF3 floxed (Jackson Labs Strain #:036260) mouse founder lines were

used[34]. For loss-of-function studies, we crossed *Irf3^flox* mice with AgRP-IRES-Cre mice (Jackson Labs Strain #012899)[43] to generate AgRP neuron-deficient IRF3 mice (AgI3KO). For gain-of-function studies, we crossed *IRF3^2D* mice with AgRP-IRES-Cre mice to generate mice expressing constitutively active IRF3 in their AgRP neurons (AgI3-2D). We crossed transgenic Nuclear tagging and Translating Ribosome Affinity Purification (NuTRAP; Jackson labs Strain #:029899) mice with AgRP-IRES-Cre mice to generate the NuTRAP^AgRP mouse line, from which we could isolate AgRP neuron-specific mRNA and nuclei. Male mice, maintained on a C57Bl/6 J background, were used for all studies.

**Other sources.** Global IRF3 knockout (IRF3KO) mice were obtained from the RIKEN BRC Experimental Animal Division (RBRC00858)[44]. C57BL/6 J were purchased from Jackson Labs (WT, Jackson Labs Strain 000664). POMC-IRES-Cre[45] were gifted by Bradford Lowell (BIDMC and Harvard Medical School).

### Animals: standard fed, fasted, and leptin-treated comparison
All animal experiments were performed with approval from the Institutional Animal Care and Use Committees of The Harvard Center for Comparative Medicine and Beth Israel Deaconess Medical Center (IACUC protocol numbers 056-2017, 024-2020, 018-2023). 6-to-11-week-old male C57BL/6 J NuTRAP^AgRP mice were fed a standard chow diet *ad libitum*. At least one day before the experiment, mice were singly-housed in a cage with wood chip bedding and handled by the experimenter using a cupping method shown to reduce anxiety in mice[46]. Mice were either maintained on their chow diet (Lab Diet, cat no 5008; fed mice), or fasted (fasted mice) overnight for 18-20 hours. At zeitgeber ZT time 2-4, mice were intraperitoneally (i.p.) injected with either vehicle (PBS) or leptin (5 mg/kg) and euthanized 3 hours later. As described in Campbell et al., 2017, with minor alterations, brains were rapidly extracted, and then placed ventral surface up into a chilled stainless steel brain matrix (catalog no. SA-2165, Roboz Surgical Instrument Co., Gaithersburg, MD) embedded in ice-cold PBS. Using GFP fluorescence to demarcate the ARCs rostral and caudal boundaries, brains were blocked to obtain a single coronal section containing the entire GFP+ arcuate, ~2 mm thick. The ARC was crudely microdissected using a surgical blade at its visually approximated dorsolateral borders and immediately snap frozen on dry ice and stored at −80 °C.

### Immunohistochemistry
Immunohistochemistry was performed as described (Campbell et al., 2017) with minor alterations. Mice were terminally anesthetized with 7% chloral hydrate (350 mg/kg) diluted in isotonic saline and transcardially perfused with phosphate-buffered saline (PBS) followed by 10% PFA. Brains were removed, stored in the same fixative overnight, and then transferred into 20% sucrose at 4 °C overnight and cut into 40-μm coronal sections on a freezing microtome.

Brain sections were washed 3 times in PBS for 10 minutes each at room temperature (RT). Sections were then subjected to antigen retrieval first by being pretreated with 1% NaOH and 1% $H_2O_2$, in $H_2O$, for 20 minutes. Sections were then treated with 0.3% glycine in $H_2O$ for 10 minutes, followed by incubation with 0.03% sodium dodecyl sulfate for 10 minutes. Sections were then blocked for 1 hour with 3% normal goat serum in PBS/0.25% Triton-X-100. 1:250 rabbit anti-phospho-STAT3 (Tyr705) (D3A7) XP (catalog #9145 S, Cell Signaling) or rabbit anti-IRF3 (catalog #4302 S, Cell Signaling) was then added and incubated overnight at 4 °C. The following day, sections were washed 3 times for 10 min in PBS at RT. Next, sections were treated with Alexa Fluor 647−conjugated donkey anti-rabbit (for pSTAT3 experiments, diluted 1:1000; catalog no. A-31573, Thermo Fisher Scientific) for 2 h in the dark at RT. Sections were washed three times in PBS, mounted onto gelatin-coated slides (Southern Biotech), cover slipped with Vectashield Anti-fade Mounting Medium with DAPI (Vector Labs,

Burlingame, CA) and sealed with nail polish. Fluorescence images were captured with an Olympus VS120 slide scanner microscope and with a confocal microscope (Zeiss LSM510 Upright Confocal System).

### Real-time PCR analysis
Cells or tissues were collected in Trizol reagent (Thermo Fisher), and tissues were mechanically homogenized using the TissueLyser II bead-mill (Qiagen). Total RNA was isolated using E.Z.N.A. Total RNA Kit II (Omega Bio-Tek) based on the manufacturer's protocol. Up to 1 μg RNA was reverse-transcribed using the High-Capacity cDNA Reverse Transcription Kit (Thermo Fisher Scientific). Using 0.5 ng cDNA in a commercial SYBR Green PCR Master Mix (Thermo Fisher Scientific) and specific gene primers (see Supplementary Table 1), qRT-PCR was performed on the QuantStudio 6 Flex Real-Time PCR System (Thermo Fisher Scientific) and normalized to the housekeeping gene *Hprt* for murine studies. Analysis of qPCR data was conducted via the $2^{-\Delta\Delta CT}$ method[47].

### Translating ribosome affinity purification (TRAP) immunoprecipitation (IP)
TRAP IP was performed as previously described (Roh et al., 2017). Briefly, for each sample 4 frozen ARC tissue samples from NuTRAP^AgRP mice between 6 and 11 weeks of age were pooled and lysed with Dounce homogenizers in low-salt IP buffer (50 mM Tris [pH7.5], 12 mM MgCl2, 100 mM KCl, 1% NP-40; 100 μg/ml cycloheximide, 2 mM DTT, 0.2units/ml RNasin, 1x Roche Complete EDTA-free protease inhibitor). After centrifugation, the supernatant was incubated with GFP ab290 antibody (Abcam) on a rotor, then incubated again with Dynabeads Protein G (Thermo Fisher). An internal control was also collected from the lipid-free supernatant prior to antibody incubation. The TRAP portion of the protocol took ~1.5 hours and RNase-inhibitors were used throughout the protocol to mitigate potential RNA degradation. RNA was isolated with the RNeasy Micro Kit (Qiagen) according to manufacturer's instructions, and then reverse-transcribed and analyzed by qRT-PCR as described above. TRAP-isolated RNA ( < 100 ng) was treated with the Ribo-Zero rRNA removal kit (Epicentre) to deplete ribosomal RNA and converted into double stranded cDNA using NEBNext mRNA Second Strand Synthesis Module (E6111L). cDNA was subsequently tagmented and amplified for 12 cycles by using Nextera XT DNA Library Preparation Kit (Illumina FC-131). Sequencing libraries were analyzed by Qubit and Agilent Bioanalyzer, pooled at a final concentration of 12pM, and sequenced on a NextSeq500.

### RNA-seq analysis
RNA-seq data was aligned using HISAT2 (Kim et al., 2015). Reads were assigned to genes using feature Counts and a GRCm38 genome modified to minimize overlapping transcripts (Liao et al., 2014). Differential expression analysis of the data was performed with edgeR using a quasi-likelihood GLM fitted to the "treatment"/condition (fed/fasted/leptin) and testing the contrasts between the groups fasted-fed and leptin-fasted using the 'QLFtest' function[48]. Significantly different genes were required to have an average expression, across group, of > 1 cpm, a fold change (FC) > 0.5, an adjusted p-value, false discovery rate (fdr), value of <0.05. Gene set enrichment analysis (GSEA) gene ontology (GO) for biological process was carried out with WebGestalt (WEB-based GEne SeT AnaLysis Toolkit)[49]. Genes with cpm <1 were omitted from the GSEA analysis.

### Isolation of AgRP neuronal nuclei from NuTRAP Mice
AgRP neuronal nuclei were isolated as previously described (Roh et al., 2017), with minor alterations. Briefly, dissected ARCs from 6-11-week-old mice were snap frozen and stored at −80C. Isolated ARCs were dounce homogenized in nuclear preparation buffer (NPB; 10 mM HEPES [pH 7.5], 1.5 mM MgCl2, 10 mM KCl, 250 mM sucrose, 0.1% NP-40, and 0.2 mM DTT), and homogenates were filtered through a

100 μM strainer and centrifuged to pellet the nuclei. Nuclei were washed with NPB, re-suspended in nuclear sorting buffer (10 mM Tris [pH 7.5], 40 mM NaCl, 90 mM KCl, 2 mM EDTA, 0.5 mM EGTA, 0.1% NP-40, 0.2 mM DTT), and filtered again through a 40 μM strainer. Isolated nuclei were sorted by flow cytometry based on AgRP neuron-specific GFP expression. With this approach we routinely obtain ~2000 AgRP neuronal nuclei per mouse.

## Nuclei processing and library preparation for ATAC-seq
NuTRAP^AgRP ARC nuclei from 4 pooled mice, per samples, were isolated, subjected to FANS, and 10,000 nuclei were collected into 500-750 uL PBS (0.1% NP40) in 1.5 mL microcentrifuge tubes and stored on ice. Two biological replicates, of 10,000 nuclei each, were used for each experimental condition. Tubes were spun at 1000 rpm for 10 mins, at 4 °C. Supernatant was removed via gentle decantation. Nuclei pellets were subjected to tagmentation after resuspension in 50ul of transposase reaction mix: 25 ul TD buffer, 2.5 ul TN5 transposase, 22.5 ul dH$_2$0. Nuclei were incubated at 37 °C with shaking at 600 rpm for 30 mins, then placed on ice. DNA was purified using a Qiagen minelute PCR purification kit and eluted in 10 ul of Qiagen Elution Buffer. Library construction was conducted using an initial PCR amplification using 10 ul of transposed DNA, 11ul H$_2$O, 2ul of 25uM Primer 1, 2ul of Primer 2, and 25 ul of NEBNext HF 2X PCR mix. The following initial amplification cycle was used: 1 cycle or 72 °C (5 min), 98 °C (30 s), 5 cycles of 98 °C (10 s), 63 C (30 s), 72 °C (1 min), hold at 4 °C. A side PCR reaction was conducted to determine the optimal number of amplification cycles used during the library construction, as previously described[50]. Briefly, the side PCR reaction was performed using 5 ul PCR amplified DNA, 4.41 ul H$_2$O, 0.25ul Primer 1 Adapter, 0.25ul Primer 2 Adapter 2 (barcoded), 0.09 ul 100x SYBR Green (Diluted from 10,000X stock with H$_2$O), 5 ul NEBNext HF 2X PCR mix. The following initial amplification cycle was used: 1 cycle of 98 °C (30 s), 20 cycles of 98 °C (10 s), 63 °C (30 s), 72 °C (1 min). Cycle vs. Linear Rn was plotted, the maximal fluorescence intensity (Rn) was determined, and the closest cycle # that produced the 1/3 Max Rn value was used to determine the additional cycles needed during library construction[50]. The required additional number of cycles (6 for all libraries) was completed by adding the sample back to the thermocycler in its same tube and master mix. Amplified libraries were purified with a Qiagen Minlute PCR purification kit, eluted in 20ul, and the volume was brought up to 100 ul with additional EB. The eluent was subjected to a two-phase size selection using Ampure Beads, initially .55X volume of elutant (55ul) was used, followed by 1.5X volume of initial sample volume (100ul x 1.5 = 150ul) minus the existing PEG that is already in the sample (150ul – 55ul = 95 ul). Finally, the library was eluted using 50ul of EB. Libraries were sequenced on a Next-seq 500 (Illumina).

## ATAC-seq data processing
Library reads were mapped to the UCSC build mm10 assembly of the mouse genome using Bowtie2, with peak calling using MACS2. ATAC-seq peaks were visualized in the WashU Epigenome Browser. Coverage summation and additional data parsing were carried out using Bedtools. Differential binding analysis, including normalization and quantification, was done in edgeR. Differential peaks were defined at log2 fold change ≥ 0.5 and a false discovery rate of <0.05. ATAC-seq peaks with a CPM < 1 were excluded from the differential analysis.

## Motif enrichment analysis and TF prioritization
Transcriptionally concordant ATAC-seq peaks were subjected to Motif Enrichment was determined using the MEME-suite Analysis of Motif Enrichment (AME) (Version 5.1.0) sequence analysis tool (Fig. 3a)[24]. We restricted our analysis to significant concordant peaks for each of the four significant ATAC-seq peak patterns (i.e., fasted-opened, fasted-closed, leptin opened, leptin-closed); genes without an annotated TSS

were removed from our analysis. AME was conducted 6 distinct times using one of the 6 possible background control sets of peaks during each analysis: (1) neutral (i.e., unchanging) peaks anywhere in the genome; (2) neutral peaks near neutral genes; (3) neutral peaks near significant genes; (4) neutral peaks near any TSS; (5) all non-concordant ATAC-seq peaks for a given conditions (e.g., leptin-opened); or (6) concordant ATAC-seq peaks that are shuffled (e.g., leptin-opened peak sequences that are shuffled). Thus, in all, 6 distinct instances of AME were performed for each peak set. We next employed a rule whereby we further considered motifs only if they were significantly enriched in ≥5 out of 6 AME instances (Fig. S3a, e).

For AME analysis, p-value is the optimal enrichment *p*-value of the motif according to the statistical test, adjusted for multiple tests using a Bonferroni correction. To determine the representative log transformed p-values (for plotting purposes), for those "enriched" motifs (at least 5 out of 6 AME instances), the minimal (most significant) p-value out of the 6 motif instances was determined, while omitting from the analysis any non-significant p-values, for those Motifs that have a single non-significant motif but are retained. Conversely, to determine the representative *p*-value for those non-enriched motifs (violate the at least 5 out of 6 significant rule) we calculated the minimal (most significant) *p*-value, of the non-significant ones. We then log-transformed the resulting *p* values (log$_2$). For the transcriptionally concordant fasted-opened condition only, p-values for 12 of the 356 TF motifs were less than e^−300 and thus could not be log transformed (ATF3.0.A, BACH1.0.C, BACH2.0.A, BATF.0.A, CRX.0.A, FOS.0.A, FOSB.0.A, FOSL1.0.A, FOSL2.0.A, JUN.0.A, JUNB.0.A, JUND.0.A). Therefore, for these motifs, only in fasted-opened condition, the log transformed p-value was determined by imputation. Evidently, none of these 12 TF motifs were qualified (due to enrichment in disqualifying chromatin conditions) to be plotted in Fig. 3g or i. All plots were made using Graphpad PRISM software (Version 9).

## In vitro leptin-induced pSTAT3 measurement experiments
Immortalized hypothalamic GT1-7 cells[37] were plated in DMEM with high glucose (Sigma, D5796), Penicillin-Streptomycin, and 10% fetal bovine serum. Cells were transiently transfected using Lipofectamine 3000 Reagent (Thermo Fisher) based on manufacturer's instructions. Briefly, dsDNA-lipid complexes were prepared using 0.5ug of plasmid expressing the murine long-form leptin receptor (OB-Rb; LepR)[51], or both LepR plasmid and the GFP-tagged IRF3 plasmid, and Lipofectamine 3000 with P1000 reagent in Opti-MEM Medium (Thermo Fisher), and then added dropwise onto cells in medium. Cells were washed with PBS, treated with serum-free DMEM, and treated with either leptin (100 nM) or vehicle (control) for 15 minutes.

## Immunoblotting
Lysis of frozen tissue was performed in 1X RIPA lysis buffer (Millipore Sigma) with the addition of protease and phosphatase inhibitors (Millipore Sigma). Tissues were mechanically homogenized using the TissueLyser II bead-mill (Qiagen). Total protein was quantified via BCA Protein Assay Kit (Thermo Scientific), protein lysate was prepared in 1X SDS in RIPA lysis buffer, and then boiled. Lysis of cells was performed as previously described (Silva et al., 2018). Briefly, media was aspirated from plated cells, and then 1X SDS in RIPA lysis buffer was added onto the cells, which were then collected, flash frozen, and boiled. The prepared protein samples were separated on Tris-HCl protein gels (Bio-Rad) and transferred to PVDF membranes (Thermo Fisher), followed by blocking in 5% blotting-grade milk (Bio-Rad). Blots were incubated in appropriate primary antibodies (1:1000 dilution unless otherwise indicated) at 4 °C overnight, and then in secondary HRP-conjugated antibody (Cytiva) for one hour at RT. Blots were developed with chemiluminescent ECL reagents (Thermo Fisher) and imaged with ImageLab™ Touch Software (Bio-Rad) on the ChemiDoc Touch Imaging System (Bio-Rad).

### In vitro IRF3 live-cell imaging and quantification

80,000 GT1-7 cells were plated in two wells of an 8-well Nunc™ Lab-Tek™ II Chambered Coverglass (Catalog # 155360) in media. The following day cells were transfected with plasmids expressing mouse LepR (OB-Rb)[51] and IRF-GFP, as described above. The following afternoon, cells were washed with 37 °C PBS and serum-free DMEM (Sigma #D1145) was added to cells prior to imaging approximately 1 hour later. Ideal cells, for which the nucleus was well demarcated, and for which the IRF3-GFP expression was apparent, were targeted and images were scanned and recorded every 10 minutes obtained using a Zeiss LSM 880 upright laser scanning confocal microscope in 3 × 3 tile-scan mode with a Plan-Apochromat 20×/0.8 M27 objective. Given the absence of a nuclear dye in this experiment, the nuclear domain was demarcated based on the clear contrast between cytoplasm and nucleus present at the 3-hour time point. Mean Fluorescence Intensity of each targeted cell's cytoplasm and nuclei was determined using ImageJ software (NIH) to enable calculation of the nuclear-to-cytoplasm (N/C) GFP intensity ratio.

### Flow cytometry to measure nuclear IRF3 in vivo

6-to-11-week-old male C57BL/6 J male mice were maintained on a standard chow diet *ad libitum*. Mice were handled for 5-minutes on the days prior to the experiment. Mice were fasted overnight for 19 h, before being injected I.P. with leptin (5 mg/kg). Using a modified version of the FAST-FIN protocol[52], 5 h later, mice were rapidly perfused with 10% formalin for 1-minute, and whole hypothalami were isolated and snap-frozen in a microcentrifuge tube on dry ice. Whole mouse hypothalamus was rapidly isolated 5 h later. Later, individual mouse hypothalami were subjected to dounce homogenization and immediately fixed using 1% formaldehyde for 7-minutes at RT, followed by quenching with 125 mM glycine for 5-minutes. Nuclei were collected and resuspended in 22% OptiPrep which was layered on top of a 43% Optiprep, and centrifuged at 10,000 g for 30-minutes. Nuclei were resuspended with Blocking Buffer (without Triton-X, in PBS) and centrifuged for 10,000 g for 10 min at 4 C. Supernatant was removed and replaced with 500 ul premade Blocking Buffer with Triton-X in PBS and nuclei were blocked for 10 min at RT. Antibody was added (IRF3, 1:250; pSTAT3, 1:250) and incubated for 1-hour. Nuclei were washed with 1X wash buffer with triton-X, resuspended in Blocking Buffer with triton-X, and incubated with 1:500 secondary antibody (Alexa Fluor 647 Donkey anti-rabbit IgG) for 30-minutes at RT while light protected. Nuclei were then spun down (600 g for 3-minutes at 4 °C) and gently resuspended in 500 ul FANS buffer with 1ul Hoechst, while in a 5 mL round-bottom FACS tube and ran on the CytoFLEX LX.

### Fasting-refeeding studies

In this study, 6-to-11-week-old male C57BL/6 J NuTRAP[AgRP] male mice were fed standard chow diet *ad libitum*. At least one day before the experiment, mice were singly housed and handled by the experimenter using a cupping method shown to reduce anxiety[46]. Depending on the experiment, mice were fasted overnight starting at ZT 8-10. The following day, between ZT 3-5, mice were injected i.p. with leptin, and 30 minutes later a weighed food grate was given. Food was measured during a 24-hour period. For the AgI3KO fasting-refeeding experiment, a white paper mesh (Alpha pads) was added to the cage in place of bedding, to allow for the accounting of spilled food when calculating food intake. Food intake was obtained by subtracting remaining food, including any spilled food in cages, from the previously weighed food amount[53]. Food intake adjusted for body weight was calculated by dividing the absolute food intake by body weight and multiplying the result by the average body weight of all mice[54].

### Telemetry experiments

All surgeries were performed in sterile conditions. Mice were anesthetized with a mix of ketamine/xylazine (100 and 10 mg/kg, respectively, IP) with additional doses of 10% of the initial dose throughout surgery to eliminate the withdrawal reflex. Mice were implanted with a radiotelemetry temperature and locomotion sensor (TA-F10, DSI) in the intraperitoneal space via laparotomy[55]. Meloxicam treatment, for analgesia, was administered prior to surgery, and then again 24hrs later. Mice were allowed to recover at least 10 days prior to experimentation. Following recovery, mice used for experiments showed no signs of discomfort and gained weight normally[55]. Core body temperature and locomotion were recorded using the radio-telemetry DSI system. The signal was sent from the telemetry probes previously implanted to receivers and converted using the PhysioTel HD and PhysioTel (DSI) hardware, which provides the mean of Tc every 5 min. All animals had at least 48 hours of acclimation in the recording chamber before the baseline was recorded. To account for occasional technical artifacts associated with continuously recording core body temperature, those 5-min interval average LMA scores that were more or less than the grand average LMA score (across all mice, conditions, and time points) plus or minus 2 times the standard deviation (i.e., 36.897 ± (2 * 2.316)), were omitted (e.g., greater than 41.5°C and less than 32.3°C). Two AgI3KO mice were removed from the leptin-treat phase of the experiment, due to their cannula being damaged, and rendered unusable, during the final leptin-treatment phase of the experiment.

### General In vivo Stereotaxic Injections

Mice were anesthetized with xylazine (10 mg/kg, i.p.) and ketamine (100 mg/kg, i.p.) and placed in a stereotaxic apparatus after assuring proper anesthesia effects with a toe pinch or tail pull. The fur around the skull was shaved and the skin sanitized. An incision in the skin overlying the skull was made and the skull surface exposed. A small hole was then drilled into the skull to expose the brain surface. The ARC was bilaterally localized according the coordinates listed in mouse brain in stereotaxic coordinates by Paxinos[56] and the tip of the nanoject injector connected to a sterile glass pipette was be lowered into the ARC bilaterally (coordinates for ARC injections were anterior −1.37 mm, lateral ± 0.3 mm, and ventral 5.70–5.80). pAAV-hSyn-DIO-H2B-mCherry (Addgene plasmid # 50459), was then slowly injected. Afterwards, the edges of the incision were reapposed with tissue adhesive (vetbond, n- butyl cyanoacrylate). Analgesia was given in the form of Meloxicam SR (4 mg/kg). Animals were allowed to recover from surgery for 2 weeks and their body weight and health conditions were closely monitored during recovery. Coordinates and injection volume used in the studies were: the ARC (anterior-posterior (AP): −1.40 mm, dorsal-ventral (DV): −5.80 mm, left-right (LR): ± 0.30 mm, 150 nl/side). Accuracy of injection was determined by the expression of mCherry driven by AgRP-IRES-Cre exclusively within the ARC, both during tissue collection and during FACS.

### Quantification and statistical analysis

Sample size, mean, and significance P values are indicated in the text, figure legends, or Method Details. Error bars in the experiments represent standard deviation (STD) from either independent experiments or independent samples. Statistical analyzes were performed using GraphPad Prism. Detailed information about statistical methods is specified in figure legends or Method Details.

### Reporting summary

Further information on research design is available in the Nature Portfolio Reporting Summary linked to this article.

## Data availability

All raw and processed RNA-Seq and ATAC-seq data has been deposited in the NCBI Gene Expression Omnibus (GEO) and assigned an accession number GEO: GSE240484. The reference genome used for RNA-

Seq and ATAC-seq raw sequencing read alignment is mm10. Source data are provided with this paper.

## Code availability

All scripts used in the current study are available on Zenodo [https://doi.org/10.5281/zenodo.10967242], the used packages are listed in the Methods session.

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

## Acknowledgements

This work was supported by an NIH 3 T32 DK 7516-32 to F.D.H., American Heart Association Postdoctoral Fellowship 18POST33990061 to F.D.H., American Diabetes Association Minority Postdoctoral Fellowship 1-19-PMF-008 to F.D.H., a Burroughs Wellcome Fund Postdoctoral Enrichment Program (PDEP) grant to F.D.H., a Nutrition and Obesity Research Center of Harvard (NORCH, # DK 040561) URM Pilot and Feasibility Grant to F.D.H., a Boston Nutrition and Obesity Research Center URM Pilot and Feasibility Grant to F.D.H, a Brain and Behavior Research Foundation Young Investigator Award to F.D.H, a Simons Foundation Simons Collaboration on Plasticity and the Aging Brain salary supplement provided to F.D.H, and R01 DK085171, R01 DK102173, R01 DK102170, and R01 DK1113669 to E.D.R. We thank the BNORC Functional Genomics and Bioinformatics Core, the BIDMC Flow Cytometry Core, the BIDMC Confocal Imaging Core, and the Molecular Biology Core Facilities (DFCI). We thank Young-Bum Kim (BIDMC) and Won-Mo Yang (BIDMC) for providing GT1-7 cells and LepR (OB-Rb) plasmid. We thank Bradford Lowell (BIDMC; HMS) and Clifford Saper (BIDMC; HMS) for their helpful comments regarding this manuscript.

## Author contributions

F.D.H. conceived this study and interpreted the results of all experiments. E.D.R supervised this study. F.D.H. optimized, designed, and executed TRAP-seq experiments. F.D.H. and E.D.R. conceived of ATAC-seq studies, which F.D.H. optimized, designed, and executed with advice from E.D.R and L.T. C.J., R.I., and H.S., performed computational and bioinformatic analyses involving sequencing data. F.D.H conceived of motif filtering/prioritization scheme. C.J. and F.D.H curated RNA-seq, ATAC-seq, and Motif enrichment analyzes. F.D.H. performed data visualization for all experiments. F.D.H. conceived of, designed, and executed in vitro, live-cell imaging, and western blot experiments. N.L., F.D.H, and S.J.P conducted in vitro and in vivo IRF3 antibody optimizations. F.D.H. optimized, designed, and executed all in vivo immunofluorescence experiments. F.D.H., N.M., and A.U. designed and executed telemetry experiments. A.G. and T.S. assisted with animal studies. F.D.H. wrote the manuscript with major input from E.D.R. along with added input from other authors.

## Competing interests

The authors declare no competing interests.
