## [Peer Review File · Nature Communications]

AgRP neuron cis-regulatory analysis across hunger states reveals that IRF3 mediates leptin's acute effectsEditorial Note: This manuscript has been previously reviewed at another journal that is not operating a transparent peer review scheme. This document only contains reviewer comments and rebuttal letters for versions considered at *Nature Communications*.

REVIEWERS' COMMENTS

Reviewer #1 (Remarks to the Author):

I am satisfied with the authors' responses to my previous comments. This current manuscript integrates genomic screens and in vivo data to implicate IRF3 as a novel transcriptional mediator of hunger suppression in AgRP neurons. In addition, the fasting/leptin responsive loci are an important resource for the field.

Reviewer #2 (Remarks to the Author):

The authors addressed most of my comments previously. I still have one remaining comment:

Figure 4: The authors cite Figure 4d for staining: 'These cells were then treated with either vehicle or 100nM leptin prior to being subjected to live cell imaging. With this approach we observed a pronounced induction in nuclear translocation of IRF3 upon treatment with leptin at the 5-hour time point (Figure 4D).' However, Figure 4a shows this and not 4d, so reference needs to be changed and would triple check all figure references in the article to be correct. Also, adding the different time points above each plot in the figure will be helpful for the reader. Finally, as this is live imaging and they do not use nuclear markers, more text in results and discussion needs to be added on how they are sure this is nuclear translocation.

Reviewer #3 (Remarks to the Author):

I previously reviewed this manuscript for Nature Neuroscience and was satisfied with the changes the authors made to address my concerns in the revised manuscript for that journal.
Rob Waterland

RESPONSE TO REVIEWERS' COMMENTS

Reviewer #1 (Remarks to the Author):

I am satisfied with the authors' responses to my previous comments. This current manuscript integrates genomic screens and in vivo data to implicate IRF3 as a novel transcriptional mediator of hunger suppression in AgRP neurons. In addition, the fasting/leptin responsive loci are an important resource for the field.

Author response: We are happy to have satisfied the reviewer's concerns and appreciate their feedback throughout this process.

Reviewer #2 (Remarks to the Author):

The authors addressed most of my comments previously. I still have one remaining comment:

Figure 4: The authors cite Figure 4d for staining: 'These cells were then treated with either vehicle or 100nM leptin prior to being subjected to live cell imaging. With this approach we observed a pronounced induction in nuclear translocation of IRF3 upon treatment with leptin at the 5-hour time point (Figure 4D).' However, Figure 4a shows this and not 4d, so reference needs to be changed and would triple check all figure references in the article to be correct. Also, adding the different time points above each plot in the figure will be helpful for the reader. Finally, as this is live imaging and they do not use nuclear markers, more text in results and discussion needs to be added on how they are sure this is nuclear translocation.

Author response: We have corrected the figure labeling error. The different time points (0hr, 1hr, 3hr, 5hr) have been listed above the associated representative confocal images. We have addressed the point regarding the nuclear label but adding the following text to the manuscript: "Given the absence of a nuclear dye in this experiment, the nuclear domain was demarcated based on the clear contrast between cytoplasm and nucleus present at the 3-hour time point". We hope this addition satisfies the reviewer and we appreciate their helpful comments.

Reviewer #3 (Remarks to the Author):

I previously reviewed this manuscript for Nature Neuroscience and was satisfied with the changes the authors made to address my concerns in the revised manuscript for that journal.

Rob Waterland

Author response: We greatly appreciate this reviewer's supportive comment, their endorsement of our work, and their helpful feedback throughout this process.